# Generative Evolutionary Strategy for Black-box Optimization

## Abstract

Numerous scientific and technological challenges arise in the context of optimization, particularly, black-box optimization within high-dimensional spaces presents significant challenges. Recent investigations into neural network-based black-box optimization have shown promising results. However, the effectiveness of these methods in navigating high-dimensional search spaces remains limited. In this study, we propose a black-box optimization method that combines an evolutionary strategy (ES) with a generative surrogate neural network (GSN) model. This integrated model is designed to function in a complementary manner, where ES addresses the instability inherent in surrogate neural network learning associated with GSN models, and GSN improves the mutation efficiency of ES. Based on our experimental findings, this approach outperforms both classical optimization techniques and standalone GSN models.

## 1  Introduction

Black-box optimization plays a vital role in both science and technology; however, it has long been an unresolved problem particularly for high dimensional problems. While low-dimensional problems, which have dimensions lesser than 100, can be optimized easily, high-dimensional optimization problems pose more significant challenges. In specific cases, such as convex functions, classical algorithms such as Evolution Strategies (ES) [7–20] and others [4–6] can efficiently tackle high-dimensional optimization problems. However, their efficiency tends to decline rapidly when faced with general black-box problems characterized by high dimensionality and non-convexity.

Furthermore, in high-dimensional optimization problems, the number of function calls inevitably grows proportionally with the dimension size. Consequently, maintaining $\mathcal{O}(N)$ time complexity is crucial in preventing the optimization process from failing owing to rapidly increasing computation time. In this context, algorithms such as the Bayesian optimization [1–3], which exhibit non-linear complexity, are at a significant disadvantage. Conversely, neural networks offer a promising solution to this problem. The field of artificial intelligence has demonstrated their considerable benefits in managing data within high-dimensional spaces, such as images or language models, while preserving linear time complexity.

Recently, GSN-based approaches, inspired by Generative Adversarial Networks (GANs) [30–32], have emerged to tackle the black-box optimization problem, offering a novel solution for high-dimensional, non-convex problems. However, in contrast to GANs, Generative Surrogate Neural networks (GSNs) face a significant challenge with learning stability in the surrogate model, and the performance of GSN-like algorithms remains limited to just hundreds of dimensions.

Hence, addressing the training instability problem is crucial for enhancing the performance of GSN-based algorithms[21, 29]. Our research aligns with this perspective, and in this work, we introduce

Submitted to 37th Conference on Neural Information Processing Systems (NeurIPS 2023). Do not distribute.

a method called Generative Evolutionary Optimization (GEO). GEO arises from the cooperative interaction between two linear complexity algorithms, ES and GSN. Furthermore, ES contributes to the stability of the surrogate network training for GSN while GSN enhances the mutation efficiency of ES, leading to their complementary functioning.

In this study, we designed an algorithm to accomplish five goals: optimizing non-convex, high-dimensional, multi-objective, and stochastic target functions, while preserving $\mathcal{O}(N)$ complexity.

In the following chapters, we explore GEO's design and the methodologies used in addressing its most significant challenge: training instability. The Related works chapter discusses two prior works, Local Generative Surrogate Optimization (L-GSO) [21] and Evolutionary Generative Adversarial Network (EGAN) [22], which serve as the foundation for GEO's core concepts. In addition, emphasis on the importance of the generator network is also discussed. The Methods chapter, we combines ideas from L-GSO and EGAN studies to clarify the aspects of GSN that require improvement and how they can be addressed. The Results chapter presents the findings from the test function experiments. Because GEO is a combination of ES and GSN, we assume a close relationship with ES, and therefore compare it against non-convex test functions commonly used in ES, such as ZDT [42], Styblinski-Tang [41], Rosenbrock [39, 40], Rastrigin [35–37], and Ackley [38] test functions. The experimental results show that GEO outperforms traditional ES and GSN as dimensionality increases, thus enabling optimizations of approximately 10,000 dimensions.

As mentioned earlier, we excluded non-linear algorithms from the comparison because our aim is to maintain $\mathcal{O}(N)$ complexity. This makes a direct comparison with Bayesian optimization, another significant branch of black-box optimization, difficult. Consequently, the issue is revisited in the Conclusion chapter.

## 2   Related works

Before delving into the structure of GEO, we will first introduce some related works, focusing on GSN and GAN algorithms. By examining earlier optimization approaches that utilized neural networks, we can gain valuable insights. As GEO is inspired by an L-GSO study (a type of GSN) and an EGAN study (a type of GAN), we will provide a brief overview of both algorithms, discussing their strengths and weaknesses for a better understanding.

### 2.1   L-GSO

Local Generative Surrogate Optimization (L-GSO) is a type of GSN that approaches the problem using a local surrogate network and a local gradient. To better understand this, let's think of a situation where we need to optimize a target function $F(x)$ within an optimization space $x \in X$ and find the best point $x$. If we identify $x_0$ as the optimal point at some stage, L-GSO samples the local space around $x_0$ and calculates pairs $[x', F(x')]$, where $x' = x_0 + \epsilon$ and $|\epsilon| << 1$. It is important that $\epsilon$ is sufficiently small so that we only sample within a space close to $x_0$. From this data, L-GSO trains a surrogate network, $C$, which acts as a local surrogate model with information around $x_0$ and can generate a local gradient. After training the surrogate network, the generator network $G$ is trained using $C$. Let $p = C(G(z))$, where $x = G(z)$ and $z$ is an input seed. $G$ is trained with a loss function $\mp p$ that either increases or decreases the prediction $p$. Finally, the trained generator suggests a search point $x$, and the iterative process continues.

This way, $G$ is trained to be a generator that produces optimal (or near-optimal) points $x$, assuming that $C$ simulates the local distribution accurately enough. Meanwhile, as the data points are focused within a localized area, the surrogate network can benefit from a stable training data region, which enables accurate gradient prediction. Another advantage is that the data information is retained in the surrogate network $C$, allowing lesser amount of data generation required when predicting the local gradient at new points.

Although the approach of utilizing GSNs in L-GSO is quite innovative, it does have some limitations. The primary constraint is that it is only applicable to single-objective function problems. The optimal point $x_0$ mentioned earlier is just a single point. However, if the optimal points (the Pareto front) consist of multiple points, the distance between these optimal points and the data sampling space (which the surrogate network must learn) will no longer be local. This undermines the fundamental premise of L-GSO. Hence, the challenge is that the optimal point of an $n$-objective function features

a Pareto front in $n-1$ dimensions. For example, in a two-objective function, the Pareto front forms a line, typically containing infinitely many non-dominated points that are distant from one another. Consequently, the locality of L-GSO becomes unsuitable for $n$-objective problems that require non-dominated sorting [58, 59].

The second problem with this algorithm is that its performance may be significantly influenced by the interaction between the hyperparameters and the test function. Specifically, the combination of the sampling hypercube size $\epsilon$ and the test function characteristics can significantly impact the algorithm's optimization performance. For instance, if the test function is convex, estimating the local gradient with any $\epsilon$ is generally not a problem; however, this can often be challenging for non-convex functions. Around the local optimum, if $\epsilon$ is smaller than the size of the localized well, the algorithm can hardly escape the local optimum. Conversely, if $\epsilon$ is too large, the local gradient cannot be accurately estimated. The interplay among the type of test function, the location of the local optimum point, and $\epsilon$ is substantial, making it difficult to determine the appropriate value of $\epsilon$ for the test function in advance; thus posing a disadvantage for black-box problems.

Conclusively, L-GSO effectively handles high-dimensional spaces using surrogate networks and improves the stability of surrogate network training through locality. However, it is clear that non-dominated sorting for multi-objective problems is not feasible, and the relationship between the hyperparameter $\epsilon$ and the test function might be too strong.

## 2.2 EGAN

The Evolutionary Generative Adversarial Network (EGAN) integrates ES to improve GAN performance. Although this method does not focus on black-box optimization, its algorithmic structure is similar to GEO.

Usually, GANs are trained with one generator and one discriminator (critic) network, which alternate. In EGAN, a scenario is considered where there is only one discriminator network; however, with multiple generator networks. The main idea of this study is to rank the generator networks using an evolutionary strategy and keep only the suitable networks. By using the prediction of discriminator $C$ as the fitness score, the generators $G$ that can increase $C(G(z))$ at each iteration survive. This process incorporates ES into the GAN, and it has been established that the introduction of ES reduces mode collapsing, which is a common issue in GANs.

Although EGAN is not directly related to optimization problems, we can gain valuable insights into improving GSN based on EGAN's concepts. Because the generator and surrogate network structure in GSN is similar to the generator and discriminator network structure in GAN, we can adopt EGAN's strategy in GSN. This forms the core basis of our research, GEO. However, the main difference between the two algorithms is that GAN operates without evolution, whereas GSN typically diverges and fails when it consists only of a single generator and a surrogate network. In the Methods section, we discuss how the working mechanism of GSN can change owing to the introduction of ES.

## 2.3 GLOnet

Global Optimization of dielectric metasurfaces using a physics-driven neural network (GLOnet) [23–27] is a study to optimize devices in electromagnetic systems. The study investigates the optimization of device structures within a specific electromagnetic system using neural network techniques.

In this study, $x$ serves as device design parameters, and the goal is to find the optimal value of $x$ that maximizes the objective value $F(x)$ of the simulator $F$. The algorithm finds $x = G(z)$, where $z$ is the input seed, and the generator $G$ is trained by backpropagating the gradient from the simulator. Technically, this is not a black-box optimization, as it receives the analytic gradient information directly from the simulator. Therefore, a surrogate network is not required.

An important implication of the GLOnet study is the necessity for a deep generator network. One might assume that since we have a gradient, we can optimize x through direct gradient descent. However, this study demonstrates that employing a generator network is more advantageous than updating x directly without a generator. In fact, this also holds true for GSN studies. The importance of deep generators is not emphasized in many GSN studies; however, it can be argued that it is omitted because they are inspired by GANs, and the significance of a deep generator is implicitly understood.

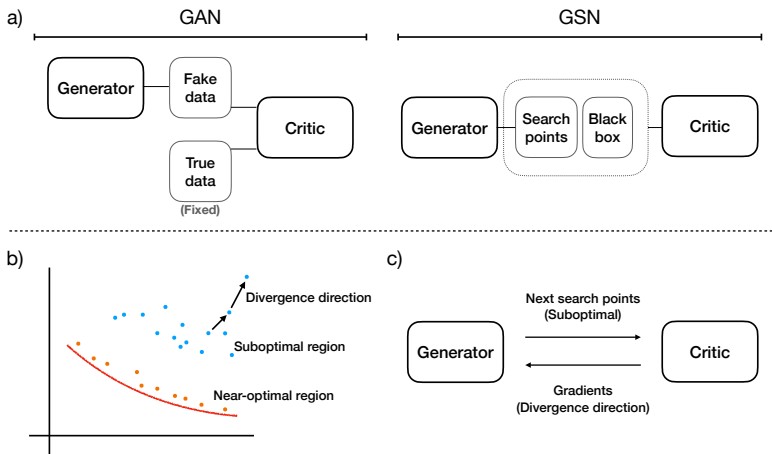

Figure 1: a) Structural differences between GAN and GSN. b) A schematic figure illustrating the training instability of GSN. The suboptimal point does not converge towards the near-optimal region, but rather diverges in the opposite direction. c) The vicious cycle between the generator and critic, which is the origin of divergence.

## 3 Methods

In the previous chapters, we examined L-GSO, EGAN, and GLOnet. Our goal is to adopt the surrogate network training strategy of L-GSO while enabling multi-objective optimization, which cannot be implemented in the local surrogate model, and also reducing the excessive dependency between the hyperparameters and the test functions. Meanwhile, we adopt the evolution strategy of EGAN; however, unlike GAN, GSN has an inherently unstable structure; hence, we need to consider how to address this instability. Additionally, EGAN assumes a single-objective fitness score; hence, we need to modify it to work with multi-objective functions.

### 3.1 A training instability

First, let's explain why surrogate training in GSNs is particularly unstable. The structure of GSNs is almost identical to GANs, hence it might seem that, just as GANs are highly successful in the fake data generation task, GSNs should also be successful. However, in practice, when we run a GSN model with just one generator network and one surrogate network, without employing any special tricks, we encounter a situation where they diverge to infinity almost immediately after starting, in most test cases. This is neither the desired result nor can we consider it being optimized. This is why GSN studies incorporate some kind of special trick.

In addition, the crucial difference between GANs and GSNs lies in the presence or absence of true data. GANs rely on fixed true data; for instance, if the task is to generate images of human faces, a large dataset of real human face photos is required. Discriminator networks possess a significant amount of fixed true data initially and learn additional fake data during training, providing a stable training region. Conversely, GSNs, which are used for black-box optimization, do not have true data. Consequently, we must start from scratch, with no prior information. The first Pareto front points that have been explored, and the surrounding points, might correspond to true data. However, the amount of data near the first Pareto front is insufficient because the first Pareto front in the objective problem is constantly changing and data is collected on-the-fly. To draw an analogy, the GAN represents a situation in which the target is anchored by true data, whereas the GSN represents a situation where the target is floating without an anchor.

This is why the critic network (discriminator network) in GANs and the critic network (surrogate network) in GSNs have similar learning structures; however, they yield completely different results. To summarize the erroneous learning process of GSN's critic network, we describe its workings as follows:

1. The critic network begins training with insufficient data.

2. The training data has minimal information about the Pareto front, which prevents the network from learning anything meaningful.

3. As the critic network is trained improperly, the gradient it provides fails to guide $x$ towards the Pareto front.

4. The generator, using incorrect gradient information, produces a point unrelated to the Pareto front when suggesting the next $x$.

5. The critic network is then retrained using the poorly generated $x$.

This leads to a vicious cycle between the generator and the critic network, a situation that does not occur in GANs owing to the presence of true data.

One solution to this problem is to create a data region that corresponds to the true data region found in GANs. In optimization problems, the most crucial information is concentrated at the first Pareto front. Hence, it is necessary to sample the area around the first Pareto front and use it as training data for the critic network. L-GSO addresses this issue by intensively sampling only the local region around $x_0$, representing the zero-dimensional first Pareto front, and supplying that data to the critic network. However, as mentioned in the related works section, the local sampling method cannot be applied to $n$-objective functions with $n > 1$. Consequently, we need an approach that is similar to the local sampling method and can also accommodate the multi-objective target functions.

Hence, we suggest using ES. The idea is to keep generators $G$ that produce $x$ values close to the first Pareto front and discard those that are farther away. By selecting $G$ through a non-dominated sorting process, we ensure that only $[x, G]$ pairs generating search points near the Pareto front survive at each iteration. Even if the critic network initially learns the incorrect region, as iterations progress, the probability of generating a search point $x$ near the Pareto front increases. Eventually, the critic network will have a stable training data region near the Pareto front; thus breaking the vicious cycle and achieving our desired outcome: stability in critic network training for multi-objective targets.

GEO was developed with the notion that ES complements GSN; however, this idea can also be reversed. As previously mentioned, from the ES perspective, the algorithm consists of mutation and (non-dominated) sorting based on fitness scores, with generator networks being the targets of mutation. In this case, mutation occurs through backpropagation from the critic network, which can be more efficient because the neural network has learned information about the Pareto front. Therefore, from the ES perspective, GSN serves as an auxiliary means to enhance the mutation efficiency of ES.

Conclusively, GEO functions in such a way that GSN complements ES, and ES complements GSN. By doing so, it integrates the strengths of both GSN and ES while mitigating their weaknesses, particularly effectively addressing the instability problem of GSN. Moreover, since both GSN and ES are $\mathcal{O}(N)$ algorithms, GEO is able to maintain $\mathcal{O}(N)$ complexity.

However, it is important to emphasize that both GSN and ES are exploitation-oriented algorithms in terms of exploit and explore strategies. Bayesian optimization, another significant branch of black-box optimization, offers a powerful feature by estimating uncertainty and incorporating it into the next step search. From that perspective, both ES and GSN lack the uncertainty estimation aspect, and even with the addition of supplementary exploration strategies, they may not reach the same level of robust exploration performance as the Bayesian optimization. Consequently, GEO, similar to GSN and ES, cannot guarantee global optimization even though the number of function calls $N$ approaches infinity. Nevertheless, GSN, ES, and GEO are free from the non-linear complexity problem that Bayesian optimization encounters. Hence, it is clear that Bayesian optimization and GEO have distinctly different optimization goals and conditions in which they are best applied.

## 3.2 Operation steps

The following are the operation sequences for GEO in the context of $n$-objective black-box optimization $F = (f_1, f_2, ..., f_i, ..., f_n)$. $G$: generator network, $C_i$: $i_{th}$ Critic network, $z$: input seed of generator, $x$: search point of search space $X$.

Pretraining steps:

1. Prepare a set of generators and $n$ critic networks. Each critic network predicts one corresponding objective. The networks have not been trained yet.

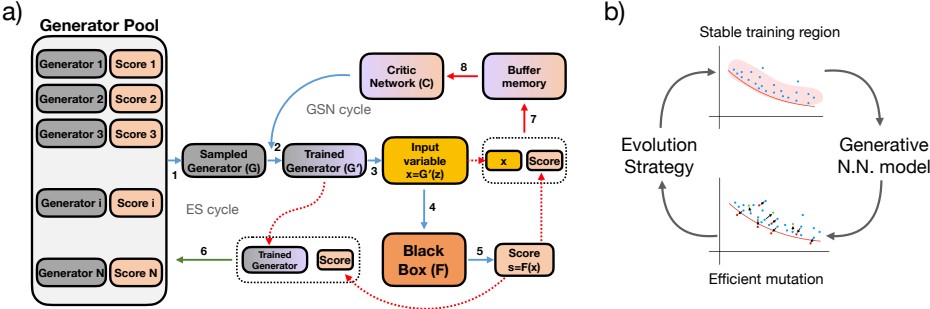

Figure 2: a) The overall algorithm of GEO. b) ES contributes to GSN by ensuring a stable training region, while GSN aids ES in carrying out efficient mutations. This creates a virtuous cycle where both algorithms complement each other's weaknesses.

2. Prepare the $[x, F(x)]$ training set for initializing the critic network. The Latin Hypercube method[52] is generally recommended as the initial sampling method for search point $x$. However, the generator network can also be initialized using weight initialization techniques for neural networks, such as Xavier or He initialization.

3. Store the initialized $[x, F(x)]$ in buffer memory, which has a maximum length.

4. Pretrain the critic network using the data stored in buffer memory. For example, $C_i$ is trained with $[x, f_i(x)]$ pairs.

Main iteration steps:

1. Randomly sample a few generators from the generator set (evolution pool).

2. Train the generators through backpropagation with the critic network $C_i$. The loss function is $-C_i(G(z))$ for the maximization problem. The training of the generators also serves as mutation from the ES perspective. Because there are $n$ critic networks, the single-objective mutation is repeated $n$ times; thus implying that for each sampled $G$, we make $n$ mutants.

3. Generate new $x' = G'(z)$ points from the mutated generators $G'$.

4. Evaluate $F(x')$ from the new $x'$ and store the pair $[x', F(x')]$ in the buffer memory. If the buffer memory's maximum length is exceeded, delete the previously stored data.

5. Train the critic networks using the data in the buffer memory.

6. Save the $[G', F(x')]$ pair back to the generator set. $F(x')$ corresponds to the multi-objective fitness score.

7. The number of generators in the evolution pool has increased with the newly stored mutated $G'$. Perform a non-dominated sort on them based on their fitness scores. Predetermine a maximum number for the generator set and retain only the generators with high fitness (top Pareto-front data).

8. Repeat the iteration.

The configuration of the critic network might have been designed to allow a single critic network to predict multiple target objectives. However, we separated the critic network independently for each objective to minimize correlation, under the assumption that it is more common for black-box problems to have independent objective targets. By dividing the critic network into $n$ parts and applying backpropagation separately, it behaves as if the single-objective problem is being run $n$ times independently. Nevertheless, it can be used for multi-objective optimization because the results are combined and ranked using non-dominated sorting.

In addition, if the target function is stochastic, age evolution can be incorporated into the non-dominated sorting step. In age evolution, we can store the time order information of generators together and remove a few of the oldest generators before performing non-dominated sorting.

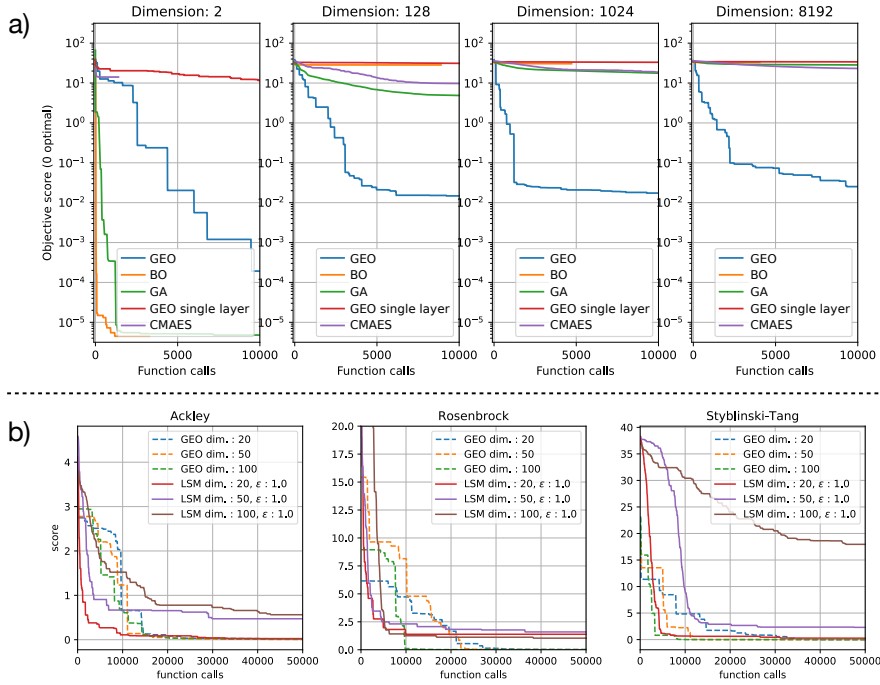

Figure 3: a) Comparison of GEO with baseline algorithms such as BO (Bayesian Optimization), GA (Genetic Algorithm), CMA-ES, and GEO with a single-layer generator. b) Comparison with LSM, a modified version of L-GSO.

The provided explanation outlines the basic algorithmic structure of GEO. As described, the operation of GEO is achieved when both the ES-direction cycle and the GSN-direction cycle work together simultaneously.

## 4 Results

Previously, we mentioned that L-GSO has a particularly strong correlation with the hyperparameter set and test function. However, all black-box optimization problems exhibit a significant correlation between the algorithm type, hyperparameter set, and target function, thus leading to entirely different results with even slight changes. Comparing the performance of optimization algorithms can be challenging precisely because of this reason. It is impossible to prove which hyperparameter set is optimal for a specific test function. Even if the optimal hyperparameter set for a particular test function is found through repeated experimentation, it would constitute overfitting to that specific test function and would no longer be considered black-box optimization. Therefore, a straightforward performance comparison between algorithms for the final results is susceptible to cherry-picking problems. Therefore, this study focuses on describing how the trend of optimization performance depends on the dimension, rather than simply comparing the values of the final results.

In the single-objective function, it is crucial to compare GEO with L-GSO. To ensure a fair comparison, we matched L-GSO's network configuration with GEO's, referring to it as Local Surrogate Model (LSM). Examining the performance changes of LSM and GEO as dimensions increase, LSM is more efficient at smaller dimensions. However, its performance declines significantly as the dimensions grow larger. This situation is also evident in classical ES and Bayesian optimization (based on Gaussian process). Although GEO is less efficient at lower dimensions, its efficiency increases as the dimensions grow, outperforming the other methods.

In the related works chapter, we discussed the importance of deep generators. To investigate this further, we conducted an experiment with GEO using a single-layer FC network as the generator. (However, the critic network remains a deep neural network.) The experiment demonstrated that the shallow layer GEO experienced a significant decrease in optimization performance.

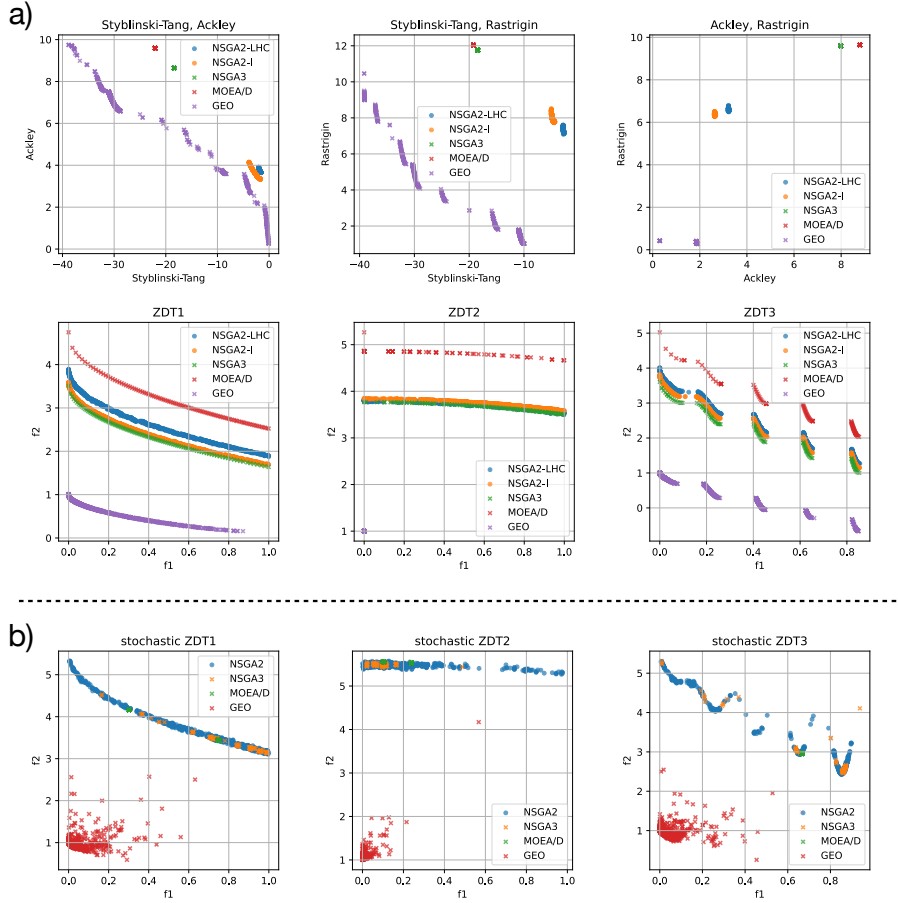

Figure 4: a) Optimization results after 100,000 function calls in two-objective function with 8192 dimensions. To investigate the influence of the initial condition, we conducted experiments differentiating between Latin Hyper Cube (LHC) initialization and point initialization (I) that is the same as the GEO neural network initial state. The results show that the influence of the initial states is insignificant. b) Optimization results in a stochastic environment with random noise added to the ZDT function, after 100,000 function calls in 8192 dimensions.

We also conducted high-dimensional experiments in multi-objective functions. According to the experimental results, as the dimension increases, the performance of the comparison group declines rapidly, whereas the performance of GEO is relatively well maintained. By the time it reaches 8192 dimensions, there is a significant difference in the final results. The comparison group tends to get trapped in local optima easily; however, GEO manages to escape local optima and makes considerable progress.

However, a limitation of GEO can be identified in one of the experimental results. The ZDT2 function has a concave Pareto-front shape. In this case, although GEO succeeds in optimization, it fails to find the entire shape of the Pareto-front and tends to collapse toward one side. We also experimented the stochastic functions by adding normal random noise to the ZDT test functions. The results for the stochastic multi-objective functions exhibit similar characteristics. Here, the performance of GEO appears to be better compared to the baseline methods; however, it also shows a similar tendency to collapse toward one side while optimizing the ZDT2 function. In the case of ZDT1 and ZDT3, although some lines of the Pareto-front are found, the collapsing tendency is stronger compared to non-stochastic functions.

Summarily, as we intended, GEO successfully overcomes the difficulties of critic network training in GSN and demonstrates stable performance. Because it does not assume locality, it shows excellent performance in multi-objective functions and operates effectively in stochastic environments.

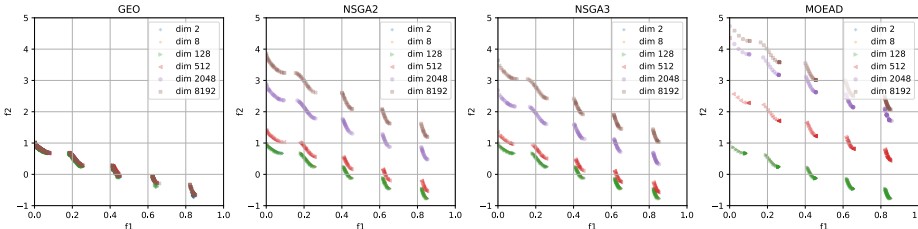

Figure 5: Changes in the optimization results of each algorithm as the dimension increases, using the ZDT3 test function after 100,000 function calls.

Although in cases with low dimensions, GEO's performance is lower compared to traditional algorithms, possibly due to too large neural network size we used; however, as the number of dimensions increases, it shows better performance than other algorithms.

## 5  Conclusion

We can observe that GEO successfully accomplishes our five primary objectives: optimizing non-convex, high-dimensional, multi-objective, and stochastic target functions while maintaining $\mathcal{O}(N)$ complexity.

In the Related works chapter, we examined insights into L-GSO and EGAN and combined them to create the foundational concept behind GEO. GSN has problems with unstable critic network training, and to address this, methods that focus on sampling around optimal values can be considered. Although L-GSO employs such an approach, introducing locality to implement it limits the algorithm to single-objective functions and results in excessive sensitivity to hyperparameters. Hence, we introduced ES combination strategy to create a stable training data region near the Pareto-front. This method can be used for multi-objective problems and also resolves the hyperparameter sensitivity problem because it does not assume a separate local $\epsilon$ size. Moreover, GSN and ES work complementarily, enhancing each other's strengths and compensating for weaknesses, leading to improved efficiency.

As indicated in the Results chapter, GEO appears to be more effective in high dimensions rather than low dimensions. For instance, ES is an algorithm that is advantageous in low dimensions when a large number of function calls is available, whereas Bayesian optimization is favorable in low dimensions with limited number of function calls. Conversely, GEO might have an advantage when many function calls are possible in high dimensions. This observation suggests that GEO has the potential to address optimization areas not covered by existing algorithms.

Because GEO is specialized for high-dimensional non-convex functions, it is worth considering its potential applications in other areas of machine learning. For example, some research trains reinforcement learning (RL) neural networks through black-box optimization [60]. As these techniques require high-dimensional black-box optimization, GEO, which specializes in high-dimensional optimization, could be a viable option.

As previously mentioned in the Results section, one difficulty in black-box optimization research could be the variability in the performance of algorithms. The performance can vary greatly depending on the combination of algorithm type, hyperparameters, and test function type. A certain algorithm and hyperparameter set might be highly effective when targeting a specific test function; however, it may yield poor results for a different test function. Hence, this can lead to biased preparation toward specific test functions, making it easier to cherry-pick results. In the worst-case scenario, one could develop an algorithm specialized for a target test function and fine-tune the algorithm through repeated experiments to obtain good results. Consequently, these results would not be considered genuine black-box optimization outcomes because they utilize prior knowledge gained through the iterative experiments. Therefore, it is challenging to determine the state-of-the-art (SOTA) status in black-box optimization research. Hence, although GEO demonstrates outstanding performance in this study, further research is necessary to determine its performance across various environments and to identify its limitations.

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
