# Supplementary Materials: Generative Evolutionary Strategy for Black-box Optimization

This appendix elaborates on the algorithm design details and additional experimental results of the Generative Evolutionary Strategy for Black-box Optimization, not covered in the main text.

## A  Model implementation details

### A.1  Neural Network

In our research, the selection and design of the neural network type plays a fundamental role as our model is grounded on a surrogate neural network. For this particular study, we used a modification of the Transformer [49] (the multi-head self-attention network) model. This structure is suitable because it can adjust to experiments of different sizes without needing any changes. Another advantage is that it can avoid a spatial correlation problem among variables.

While it is true that Convolutional Neural Networks (CNNs) and Recurrent Neural Networks (RNNs) are beneficial when it comes to dimension expansion, they do come with their share of challenges, notably the spatial correlation problems. In contrast, a problem of attention-based neural network is computational complexity, which escalates to $\mathcal{O}(d^2)$ when the variable size hits $d$. This often results in Graphics Processing Unit (GPU) memory shortages. Therefore, we had to look for other options. We could either choose a network like CNN, which has complexity $\mathcal{O}(d)$, or find a completely new solution.

To address this, we introduced a strategy we have termed the trunk-branch trick. While this method continues to employ the attention mechanism, it subtly modifies the structure to reduce complexity. First, we create a trunk network. Then, from the trunk, we attach $M$ branch networks. Each of these branches predicts a segment of the total dimension $d$ of the target variable, specifically $d/M$. The structures of the generator and critic are mirror images of each other, meaning they are symmetrical. For the generator, it starts processing in the trunk and then spreads out to the branches to create the target variable $x$. The critic works the other way around: its branches take parts of $x$ and bring them back together in the trunk.

Implementing this method effectively brings down the complexity to $\mathcal{O}(d^2/M)$, offering a solution to GPU memory shortages. However, the trunk-branch affects the optimization performance of GEO. The related experiment is described in the Additional experiments chapter.

### A.2  Non-dominated sorting

When dealing with scenarios that involve multiple target functions, there is often a competitive interaction between each function's optimal points. Imagine a variable $x$ that increases the value of one function, $f_1(x)$, while it simultaneously reduces the value of another function, $f_2(x)$, and vice versa. In such situations, we use a method known as non-dominated sorting to identify the optimal point.

Submitted to 37th Conference on Neural Information Processing Systems (NeurIPS 2023). Do not distribute.

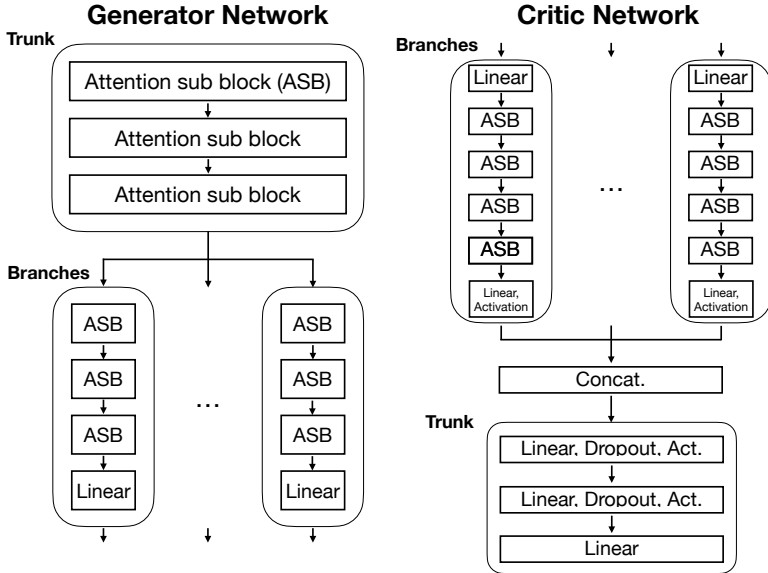

Figure 1: Generator and critic network structure.

Let's look at minimization problems to understand this better. We label a specific point, $x_0$, as non-dominated when we cannot find another point $x$ that fulfills the condition $f_i(x) < f_i(x_0)$ for all the target objective functions $f_i$. These non-dominated points form a group known as the first Pareto-front. Consequently, we can arrange these Pareto-fronts in an order, creating a sequence like the 1st Pareto-front, 2nd Pareto-front, and so on. This ordered arrangement is what we refer to as non-dominated sorting.

There are many different methods in non-dominated sorting, each with its own computational complexity. However, we did not place too much emphasis on this aspect, as our pool size was small enough that the time required for non-dominated sorting was considerably less than that required for neural network computations.

## A.3 Training details

In the Methods section of the main text, we noted the potential tendency of GEO towards the exploitation with regard to the exploit-explore strategy. This trait primarily emerges from its design, a combination of ES and GSN. Furthermore, this tendency could limit the algorithm's potential for exploration. Therefore, we added several strategies to boost its exploration abilities.

One approach we used was to try various learning rates. We prepared a wide range of learning rates, from small to large values, and applied them either randomly or all at once for each mutation event. This approach can provide an escape path if the algorithm gets stuck in a local optimum. We also considered implementing random mutations, changing the parameters of the sampled generator network or specific layers to create mutants. At the very least, these additional techniques do not compromise performance, although we could not definitively establish that they enhanced it.

We also tested various hyperparameters which directly impact experimental results. For example, the pool size is important because it plays a key role in shaping the Pareto-front.

For certain functions, such as the ZDT functions, it is necessary to set boundaries. The boundary construction method can affect performance. We can find the results of these experiments in the Additional experiments section.

For the training of the generator (mutation), we chose to train in the $n$ separate directions using $n$ independent critic networks. There can be another approach where we sum up $n$ fitness scores and then increase the average score. However, this method could possibly lead to a bias towards points in the middle of the Pareto front. Moreover, this approach introduces new hyperparameters related

to the normalization of each fitness score, which significantly influences the algorithm's behavior. Therefore, to circumvent the introduction of new hyperparameters, we chose to conduct the training $n$ times independently.

# B    Experiment details

## B.1    Non-convex test functions

In our experiment, we analyzed multiple dimensions, each requiring comparative investigation. To make the analysis process consistent, we normalized all test functions in line with their dimension sizes. Most of test functions have the ground state as $0$; however, the Styblinski-Tang function does not. Therefore, we recalibrated its value to set $0$ as the ground state. Although all these functions fall under the non-convex category, the Rosenbrock function has distinct properties, as its optimal solution resides within a flat valley region

## B.2    Computational details

We aimed to run the algorithm with minimal hardware acceleration. Therefore, we designed the algorithm to be compatible with a single GPU. When we faced memory shortages, we employed the trunk-branch trick, a method we previously discussed in the neural networks chapter. By using the NVIDIA Tesla v100 32GB GPU, it took about three days to execute 100,000 function calls and trainings. The majority of this computational time was dedicated to training the neural networks.

We could enhance the efficiency of neural networks by incorporating a CNN or optimizing the attention model. However, given the scope of our task, which involved 100,000 function calls across 8192 dimensions, the model we currently have is sufficient. Thus, we did not devote substantial time to network optimization. Moreover, we have to avoid excessively refining the algorithm for a specific test function. Such over-tuning could lead to overfitting and consequently limit the model's performance on different problems especially for real-world problems.

In our study, we made 100,000 function calls for each task. Although this might seem like a considerable number, it is not particularly large when viewed within the context of machine learning and neural networks. Therefore, we found a smaller model that could run on a single GPU to be the most suitable choice; note that if we increase the model size, we can potentially see a decrease in performance. However, if the number of function calls were to increase significantly, we might require a larger model. This could occur in scenarios where the target black-box is readily parallelizable.

## B.3    Further investigations

As we previously noted in the main text, Bayesian optimization forms a vital branch of optimization research. This is particularly the case for the Gaussian process-based Bayesian optimization, a practical method employed in machine learning and hyperparameters optimization. Despite its utility, it is important to note that it comes with an $\mathcal{O}(N^3)$ complexity, which stands as a significant hurdle. A proposed potential solution to this issue is a neural-process. [61] This approach is receiving increasing interest because it offers Gaussian process-like uncertainty estimation while simultaneously reducing time complexity. However, it still demands the accumulation of search point data for uncertainty prediction. This requirement results in a continuous escalation in computation time, preventing it from meeting the $\mathcal{O}(N)$ complexity when employed in optimization tasks.

In the ES domain, numerous modifications have been developed based on well-known and commonly used algorithms such as GA and CMA-ES [54–56]. In reality, though, optimization experiments typically deal with around 100 dimensions. Sometimes, these experiments may handle larger dimensions; in that case however, they usually focus on simpler convex shapes such as the spherical function and Rosenbrock function; note that Rosenbrock has long flat region around the global minimum.

When it comes to experiments in the 100-1000 dimension range, GSN-based optimization studies have given promising results. [29]. However, the test functions used in these studies are often special types of test function rather than common test functions that have been widely used in ES studies, making it difficult to compare performance. In some cases, Generative Neural Network (GNN) is used, but not the surrogate model. A GNN-based study [28] has shown optimization results around 10

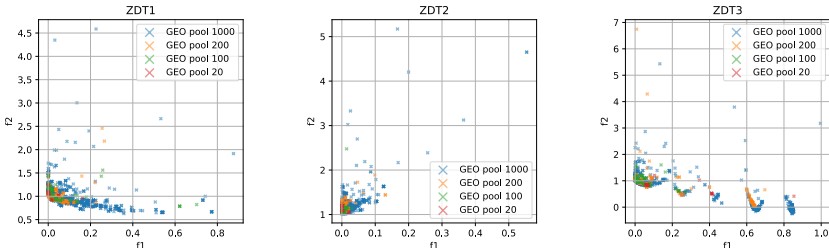

Figure 2: Pool size experiments. Optimization results in 8192 dimension after 100,000 function calls.

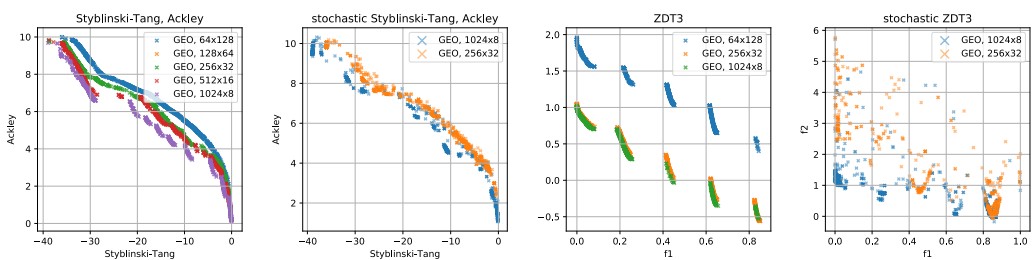

Figure 3: Trunk-branch trick experiments. Optimization results in 8192 dimension after 100,000 function calls. For $M$ branches, $8192 = (8192/M) \times M$.

dimensions; however, this is considerably distanced from our target level of dimensions. Therefore, we did not take this algorithm into account.

Research based on surrogate models is also ongoing. For instance, Pysamoo [57] offers packaged optimization algorithms based on surrogate models. Nonetheless, we found that these offered models cannot be effectively applied in high dimensions due to time complexity problems. However, as the package is regularly updated, it could overcome this problem in the future.

Finally, Particle Swarm Optimization (PSO) [5] is a different type of algorithm from ES, but it has some similar features. Here, a certain number of elements swarm towards the global optimum while preserving their group. Therefore, we can consider how to combine PSO and GSN. However, within the scope of this study, we were unable to devise a way to integrate PSO. While ES provides a simple means to link GSN's backpropagation and non-dominated sorting, establishing a similar connection in PSO poses a challenge.

## C  Additional experiments

### C.1  Pool size

The pool size has a direct impact on performance, especially if it is too small. In the experiment, a small pool GEO encounters problems when identifying the overall shape of the Pareto-front. This problem may arise when the algorithm loses some lines of the Pareto-front in non-dominated sorting, making them difficult to recover.

### C.2  Trunk-branch structure

In the neural network section, we mentioned that the trunk-branch trick is a temporary solution to address the problem of GPU memory shortage. Hence, we also explored the performance related to the trunk-branch strategy.

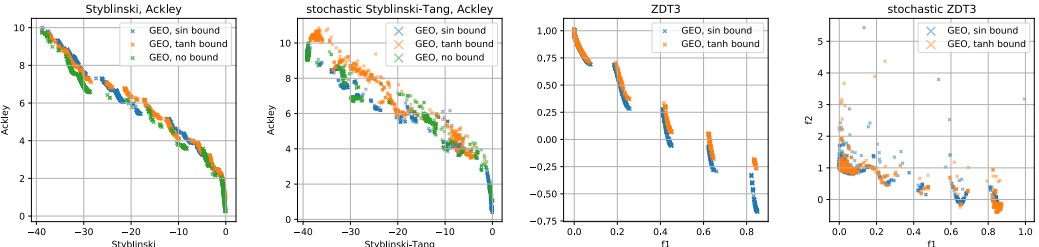

Figure 4: Boundary function experiments. Optimization results in 8192 dimension after 100,000 function calls.



Figure 5: An experiment to black-box optimization of LeNet-5 trained with MNIST. Maximum score $s = 1.0$. Although optimization is successful, it do not produce the intended smooth handwriting image.

The experimental result clearly shows that as the number of branches increases, the performance decreases. The trunk-branch structure enhances the time and space efficiency of the attention network but at the expense of optimization performance. Therefore, these results suggest that there are limits to increasing the number of branches in extremely high dimensions. It might imply the necessity for fundamental changes, such as adopting CNNs.

## C.3  Boundary conditions

ZDT test functions require boundary conditions in the search space $X$. To implement these boundary conditions, we attached an additional function at the end of the generator to enforce boundaries.

$$x = (bound_{max} - bound_{min}) \frac{B\left(G(z)\right) + 1}{2} + bound_{min}$$

Boundaries could be implemented with functions such as $B = tanh$ and $B = sin$, with the periodic boundary condition of the $sin$ function showing slightly better results. This might occur due to the use of the $tanh$ function, which could create a bias at the edges, leading to a concentration of points that exceed the boundary at the edge. This, in turn, generates redundant data during the training of the critic network. Hence, if a boundary is necessary, it is recommended to use a periodic boundary condition.

## C.4  Manifold issue

In the L-GSO research, it was suggested that the surrogate could effectively discern manifold structures, and that optimization performance would likely improve within manifold structures than without a manifold structure. As GEO employs a similar surrogate neural network-based algorithm, the same circumstances may arise.

To visually verify this, we conducted an image generation experiment. We considered a simple LeNet-5 [44], trained on the MNIST dataset, as a black-box and optimized it. If GEO prefers manifold structures, images created through black-box optimization should exhibit a smooth shape (likely resembling actual human hand-drawn images).

However, the results are contrary. Even though the optimization is successful, a smooth shape does not emerge; instead, it produces a noise image resembling an adversarial attack. This casts doubt on previous research findings suggesting that GSN-based optimization is better suited for learning manifold structures.

Table 1: Optimization results of Ackley function in low dimensions. 20,000 function calls. 10 repeats.

| | Ackley | | | |
|---|---|---|---|---|
| Dimension | 2 | 4 | 8 | 16 |
| GEO | $0.0000 \pm 0.0000$ | $0.0071 \pm 0.0076$ | $0.1009 \pm 0.0432$ | $0.3014 \pm 0.2451$ |
| GEO 1-layer | $0.0000 \pm 0.0000$ | $0.0030 \pm 0.0023$ | $0.8575 \pm 0.7513$ | $1.9020 \pm 0.2054$ |
| GA | $0.0001 \pm 0.0002$ | $0.0016 \pm 0.0008$ | $0.0073 \pm 0.0033$ | $0.0411 \pm 0.0066$ |
| CMAES | $0.0000 \pm 0.0000$ | $0.0000 \pm 0.0000$ | $0.0000 \pm 0.0000$ | $0.0000 \pm 0.0000$ |
| LSM $\epsilon1.0$ | $0.0053 \pm 0.0081$ | $0.0261 \pm 0.0118$ | $0.1056 \pm 0.0292$ | $0.2036 \pm 0.1139$ |
| LSM $\epsilon0.2$ | $0.0006 \pm 0.0003$ | $0.6754 \pm 1.3312$ | $0.0354 \pm 0.0076$ | $0.8967 \pm 0.9222$ |
| | 32 | 64 | 128 | |
| GEO | $0.0694 \pm 0.0576$ | $0.0361 \pm 0.0185$ | $0.0296 \pm 0.0136$ | |
| GEO 1-layer | $2.7931 \pm 0.1655$ | $3.4449 \pm 0.1133$ | $3.8488 \pm 0.1002$ | |
| GA | $0.1132 \pm 0.0163$ | $0.2795 \pm 0.0304$ | $0.5510 \pm 0.0369$ | |
| CMAES | $0.0000 \pm 0.0000$ | $0.0000 \pm 0.0000$ | $0.0001 \pm 0.0000$ | |
| LSM $\epsilon1.0$ | $0.2430 \pm 0.1160$ | $0.3432 \pm 0.1225$ | $0.8251 \pm 0.2573$ | |
| LSM $\epsilon0.2$ | $2.5080 \pm 1.2227$ | $3.4657 \pm 0.3694$ | $3.3817 \pm 0.3004$ | |

Table 2: Optimization results of Rosenbrock function in low dimensions. 20,000 function calls. 10 repeats.

| | Rosenbrock | | | |
|---|---|---|---|---|
| Dimension | 2 | 4 | 8 | 16 |
| GEO | $0.0000 \pm 0.0000$ | $0.0543 \pm 0.1507$ | $0.5090 \pm 0.4545$ | $0.5378 \pm 0.5966$ |
| GEO 1-layer | $0.0000 \pm 0.0000$ | $0.3062 \pm 0.2330$ | $0.8592 \pm 0.2276$ | $3.2036 \pm 1.7442$ |
| GA | $0.0001 \pm 0.0001$ | $0.0888 \pm 0.0488$ | $0.5197 \pm 0.1102$ | $0.8267 \pm 0.0755$ |
| CMAES | $0.0000 \pm 0.0000$ | $0.0000 \pm 0.0000$ | $0.0000 \pm 0.0000$ | $0.0000 \pm 0.0000$ |
| LSM $\epsilon1.0$ | $0.3805 \pm 0.3780$ | $0.8514 \pm 0.3439$ | $1.3913 \pm 0.1683$ | $1.7246 \pm 0.1924$ |
| LSM $\epsilon0.2$ | $0.5814 \pm 0.3466$ | $0.5992 \pm 0.3083$ | $0.6444 \pm 0.3757$ | $0.7882 \pm 0.2399$ |
| | 32 | 64 | 128 | |
| GEO | $0.1705 \pm 0.2817$ | $0.0564 \pm 0.0975$ | $0.0164 \pm 0.0170$ | |
| GEO 1-layer | $11.3029 \pm 1.3108$ | $32.3401 \pm 3.7944$ | $61.4009 \pm 4.9140$ | |
| GA | $1.7519 \pm 0.5589$ | $4.0404 \pm 0.4313$ | $5.9195 \pm 0.2543$ | |
| CMAES | $0.6532 \pm 0.0278$ | $0.9068 \pm 0.0150$ | $0.9734 \pm 0.0102$ | |
| LSM $\epsilon1.0$ | $1.8820 \pm 0.5490$ | $2.1025 \pm 0.7888$ | $2.0884 \pm 0.7405$ | |
| LSM $\epsilon0.2$ | $1.5623 \pm 0.7663$ | $17.4755 \pm 25.1455$ | $38.9798 \pm 23.4691$ | |

Table 3: Optimization results of Rastrigin function in low dimensions. 20,000 function calls. 10 repeats.

| | Rastrigin | | | |
|---|---|---|---|---|
| Dimension | 2 | 4 | 8 | 16 |
| GEO | $0.0000 \pm 0.0000$ | $0.1027 \pm 0.1635$ | $0.1434 \pm 0.2483$ | $0.8459 \pm 0.6891$ |
| GEO 1-layer | $0.0000 \pm 0.0000$ | $0.3483 \pm 0.1219$ | $0.4758 \pm 0.2170$ | $1.2868 \pm 0.3246$ |
| GA | $0.0000 \pm 0.0000$ | $0.0000 \pm 0.0000$ | $0.0010 \pm 0.0009$ | $0.0127 \pm 0.0048$ |
| CMAES | $0.2985 \pm 0.3300$ | $0.4477 \pm 0.2168$ | $0.5721 \pm 0.2021$ | $0.4166 \pm 0.1446$ |
| LSM $\epsilon1.0$ | $0.5375 \pm 0.4406$ | $3.6183 \pm 1.9796$ | $5.6259 \pm 1.2943$ | $5.5754 \pm 1.3568$ |
| LSM $\epsilon0.2$ | $0.0000 \pm 0.0000$ | $0.5076 \pm 0.4908$ | $0.3248 \pm 0.3222$ | $0.4995 \pm 0.2248$ |
| | 32 | 64 | 128 | |
| GEO | $1.8010 \pm 1.2062$ | $1.9690 \pm 1.3727$ | $0.8947 \pm 1.1321$ | |
| GEO 1-layer | $2.8961 \pm 0.3151$ | $4.9785 \pm 0.2506$ | $6.4839 \pm 0.2165$ | |
| GA | $0.1580 \pm 0.0452$ | $0.5013 \pm 0.0478$ | $0.9218 \pm 0.0550$ | |
| CMAES | $0.5006 \pm 0.1526$ | $0.5208 \pm 0.0961$ | $0.6630 \pm 0.0978$ | |
| LSM $\epsilon1.0$ | $5.1430 \pm 1.5115$ | $7.8261 \pm 0.8673$ | $8.8664 \pm 0.6537$ | |
| LSM $\epsilon0.2$ | $0.6955 \pm 0.3504$ | $4.9844 \pm 2.2791$ | $7.5184 \pm 1.6464$ | |

Table 4: Optimization results of Styblinski function in low dimensions. 20,000 function calls. 10 repeats.

| | Styblinski-Tang | | | |
|---|---|---|---|---|
| Dimension | 2 | 4 | 8 | 16 |
| GEO | $0.0000 \pm 0.0000$ | $0.0009 \pm 0.0014$ | $0.0023 \pm 0.0017$ | $0.0161 \pm 0.0176$ |
| GEO 1-layer | $2.1695 \pm 3.2087$ | $8.5127 \pm 2.2338$ | $15.9916 \pm 1.1828$ | $22.4661 \pm 0.5406$ |
| GA | $0.7068 \pm 2.1205$ | $0.7069 \pm 1.4137$ | $3.5346 \pm 2.3707$ | $5.2253 \pm 0.9198$ |
| CMAES | $7.0684 \pm 5.4751$ | $12.7231 \pm 3.2391$ | $10.9560 \pm 2.5971$ | $9.7190 \pm 2.3708$ |
| LSM $\epsilon1.0$ | $26.2230 \pm 1.9996$ | $7.6978 \pm 1.2559$ | $6.4853 \pm 3.9141$ | $9.5768 \pm 4.0347$ |
| LSM $\epsilon0.2$ | $27.6515 \pm 2.8802$ | $16.1667 \pm 1.6806$ | $12.1698 \pm 2.0510$ | $19.3622 \pm 3.1020$ |
| | 32 | 64 | 128 | |
| GEO | $0.0064 \pm 0.0092$ | $1.4138 \pm 4.2410$ | $0.0000 \pm 0.0000$ | |
| GEO 1-layer | $25.3427 \pm 0.5938$ | $27.5193 \pm 0.6898$ | $29.3574 \pm 0.3288$ | |
| GA | $7.9346 \pm 1.0712$ | $11.2910 \pm 0.5126$ | $18.3960 \pm 0.3724$ | |
| CMAES | $8.4820 \pm 2.2159$ | $9.9841 \pm 0.6612$ | $9.2883 \pm 0.5663$ | |
| LSM $\epsilon1.0$ | $9.7605 \pm 4.3111$ | $8.7600 \pm 2.9911$ | $17.5349 \pm 4.3426$ | |
| LSM $\epsilon0.2$ | $18.9152 \pm 2.6356$ | $32.2802 \pm 2.4185$ | $33.5116 \pm 2.6515$ | |

Table 5: Optimization results of test functions in high dimensions. 50,000 function calls. 10 repeats

| | Ackley | | |
|---|---|---|---|
| Dimension | 256 | 512 | 1024 |
| GEO | $0.0091 \pm 0.0036$ | $0.0117 \pm 0.0037$ | $0.0084 \pm 0.0029$ |
| GA | $0.3294 \pm 0.0219$ | $0.7342 \pm 0.0273$ | $1.4256 \pm 0.0477$ |
| CMAES | $0.0000 \pm 0.0000$ | $0.0003 \pm 0.0000$ | $0.0291 \pm 0.0035$ |
| | Rosenbrock | | |
| Dimension | 256 | 512 | 1024 |
| GEO | $0.0006 \pm 0.0006$ | $0.0004 \pm 0.0003$ | $0.0005 \pm 0.0004$ |
| GA | $5.0726 \pm 0.2486$ | $5.9915 \pm 0.2358$ | $6.9435 \pm 0.1485$ |
| CMAES | $0.9742 \pm 0.0062$ | $1.0011 \pm 0.0337$ | $1.0292 \pm 0.0301$ |
| | Rastrigin | | |
| Dimension | 256 | 512 | 1024 |
| GEO | $0.2034 \pm 0.4046$ | $0.0018 \pm 0.0020$ | $0.0034 \pm 0.0057$ |
| GA | $0.5636 \pm 0.0317$ | $0.9810 \pm 0.3457$ | $1.7443 \pm 0.0638$ |
| CMAES | $0.9573 \pm 0.1040$ | $1.3981 \pm 0.1383$ | $3.7305 \pm 0.6978$ |
| | Styblinski-Tang | | |
| Dimension | 256 | 512 | 1024 |
| GEO | $0.0000 \pm 0.0000$ | $0.0000 \pm 0.0000$ | $0.0000 \pm 0.0000$ |
| GA | $14.4536 \pm 0.3616$ | $22.3592 \pm 0.2035$ | $28.8298 \pm 0.1541$ |
| CMAES | $9.6913 \pm 0.4508$ | $9.3711 \pm 0.1878$ | $9.2048 \pm 0.2450$ |