# OpenReview forum: "Generative Evolutionary Strategy For Black-Box Optimization"
_NeurIPS.cc/2023/Conference — Submitted to NeurIPS 2023_

### Official Review · Reviewer_Hd6y · 2023-07-03

**Soundness:** 2 fair
**Presentation:** 2 fair
**Contribution:** 2 fair
**Rating:** 3
**Confidence:** 4

**Summary:**

The paper introduces a Generative Evolution Optimization (GEO) algorithm to black-box optimization, introducing. The GEO algorithm is claimed to combine the strengths of Evolution Strategy (ES) and Generative Surrogate Network (GSN) to address the limitations of Bayesian optimization and other existing methods. Some benchmark functions are tested to verify the performance of GEO.

**Strengths:**

Originality: The paper introduces GEO which combines the strengths of L-GSO and Evolutionary Generative Adversarial Networks (EGAN). I did not see such method before.
Quality: The authors provide explanations of the GEO method, including its foundational concepts, operation steps, and algorithmic structure. Some benchmark functions have been tested.
Significance: The paper addresses a significant challenge in the field of black-box optimization, e.g., the optimization of non-convex, high-dimensional, multi-objective, and stochastic target functions.


**Weaknesses:**

Some claims and concepts are not adequate, like the O(N) complexity. Without the target of finding global optimal, we can design various methods that achieve O(N) complexity easily.
Some related works are not cited adequately, like Xavier and He initialization.
The experiments seem to be limited to specific test functions. Performance on such few benchmark functions are not convincing
The paper discusses the potential application of GEO in other areas of machine learning, such as reinforcement learning. However, it does not provide any empirical evidence or case studies to support these claims. Including real-world applications or case studies could strengthen the paper's significance and practical relevance.
The paper does not clearly outline the limitations of the GEO method, which could be beneficial for future research and application of the method.
The paper mentions the tendency of GEO to collapse towards one side while optimizing certain functions, but it does not delve into why this happens or how it could be mitigated. A more in-depth analysis of this issue could improve the paper's quality.


**Questions:**

The paper mentions the tendency of GEO to collapse towards one side while optimizing certain functions. Could you provide more insight into why this happens?
The paper mentions that GEO is more effective in high dimensions rather than low dimensions. Could you provide more insight into why this is the case?
The paper discusses the potential application of GEO in other areas of machine learning, such as reinforcement learning. Could you provide any empirical evidence or case studies to support these claims?
What are the experimental settings? Why we choose those benchmark functions? What are the measurements of the performance? What are the parameter settings for the algorithms? How many extra computational costs are introduced in the optimization procedure? Those are all unclear.

**Limitations:**

Societal impact of the work is not discussed in this paper. Furthermore, limitations of the proposed technique are not discussed clearly.

---

> ### Author Rebuttal · Authors · 2023-08-02
>
> Thank you for your review and the many questions.
> I will provide answers to each of your questions in turn.
>
> Q1. The paper mentions the tendency of GEO to collapse towards one side while optimizing certain functions. Could you provide more insight into why this happens?
>
> A1. In the case of ZDT2 where collapsing occurs strongly, it has a concave shape of the Pareto-front, unlike ZDT1 (convex shape) and ZDT3 (partially convex, partially concave shape). We hypothesize that when the Pareto front is concave, the edge state's probability of being sampled in the ES process increases, leading to gradual collapsing to one side.
>
> We acknowledge that there can be simple solutions, such as setting reference axes and referencing them during the ES process to rebalance the weights. However, this would require additional hyperparameter tuning and would be considered a separate task from the original ES algorithm. If we were to suggest a GEO that includes Reference axes, it would be considered a separate algorithm from the original GEO. (e.g. In Classical ES, NSGA and R-NSGA are considered separate algorithms)
>
> This explanation was included in the earlier version, but was removed to meet the paper's length limit. We'll add it back to the supplement.
>
> Q2. The paper mentions that GEO is more effective in high dimensions rather than low dimensions. Could you provide more insight into why this is the case?
>
> A2. This is an important question. The reduced performance of GEO in lower dimensions is due to GEO being based on neural networks.
>
> Neural networks show very high generalization performance when learning from a large amount of data, but when there's less data, performance sharply decreases due to overfitting. Particularly, this tendency is stronger with larger neural network sizes.
>
> Optimization in lower dimensions can be achieved with fewer data points (e.g., optimization in 2-dim. might be sufficiently done with 100 data points), so algorithms based on neural networks may show weakness in these situations.
>
> Also, we used a specific structure called a self-attention Transformer in GEO for this experiment (We explained the reason in Supplement). Transformers are known to be more suitable for large data but tend to perform poorly with small data. This trend made GEO's performance at lower dimensions get worse. Structurally, the decline in neural networks' performance with less data is an intrinsic problem that cannot be avoided. Hence, I emphasized in the discussion section that GEO is suitable for higher dimensions when there's a possibility for many target function evaluations.
>
> The characteristics of neural networks are generally well-known, so I thought that an explanation might not be necessary. However, it seems that a more comprehensive explanation would indeed be beneficial. Adding a discussion on this topic to the supplement appears to be a good approach.
>
> Q3. The paper discusses the potential application of GEO in other areas of machine learning, such as reinforcement learning. Could you provide any empirical evidence or case studies to support these claims?
>
> A3. This is mentioned with consideration of the paper “Evolution strategies as a scalable alternative to reinforcement learning” published by OpenAI in 2017 (In the discussion section, we cited this paper). In this paper, the authors demonstrate that reinforcement-type neural net problems can be trained with ES instead of traditional RL training techniques. Despite being presented by OpenAI, this research did not gain significant attention, primarily due to its performance compared to conventional RL techniques like DQN and A3C.
> Hence, we mentioned our interest in applying GEO in a similar manner. Since GEO is a kind of ES, specialized for high dimensions, it might be worthwhile to challenge this kind of problem with GEO.
> (Note that, ES can be effectively applied to the neural network, as the neural network itself is a kind of special type of convex function)
>
> Q4. What are the experimental settings? Why we choose those benchmark functions? What are the measurements of the performance? What are the parameter settings for the algorithms? How many extra computational costs are introduced in the optimization procedure? Those are all unclear.
>
> A4. All experimental settings are explained in the Supplement. Please see the details there
>
> When we wrote the initial version of this paper, we included all the experimental setting in the 9-page main text.. This led to many confusions of readers. Thus, we allocated most of the main text pages to the explanation of GEO's working principle, and included the experimental settings and important additional experiments in the Supplement. There are so many details in Supplement, so it's challenging to put everything in the Rebuttal. We kindly ask that you refer to the Supplement.
>
> The benchmark functions we used are those traditionally used in classical optimization researches. Sometimes, a few studies (mainly based on neural networks) completely ignore traditional test functions in performance measurement, making it difficult to compare with previously developed algorithms. It is important to measure performance based on well-established and validated test functions, such as the ZDT functions.
>
> Performance measurement is based on how close we got to the global optimum relative to the number of target function evaluations. This is predicated on one of the primary assumptions in black-box optimization that the target function evaluation is the most expensive resource. Nearly all studies in this field follow this assumption.
>
> How many extra computational costs - See supplement for the issue of non-dominated sorting cost
>
> We kindly ask for your understanding that we put all experimental setting information in the Supplement.
> Due to the 9-page limit, including all experimental details would reduce the GEO's main algorithm explanation, leading to reader confusion.
>
> Once again, thank you for your earnest review

---

> > ### Comment · Reviewer_Hd6y · 2023-08-16
> > **Thank your for the response**
> >
> > Thank you for the responses. I have read the rebuttal. Though I get the idea of the answers, I still feel the statements are somewhat intuitive and thus vague. This issue also exists in the manuscript as I mentioned in the "weakness". I believe a more rigorous description would help better clarify the contribution and the technical merits.

---

> > > ### Author Response · Authors · 2023-08-20
> > > **Thank you for your insights.**
> > >
> > > Thank you for your insights. I understand your concerns about the intuitive nature and perceived vagueness in the manuscript. I will make an effort to provide a more rigorous description of the concepts and technical details in the subsequent revision.
> > >
> > >  if space is insufficient in the main text, I will provide as much information as possible in the supplementary materials.

---

### Official Review · Reviewer_7diD · 2023-07-05

**Soundness:** 4 excellent
**Presentation:** 3 good
**Contribution:** 4 excellent
**Rating:** 7
**Confidence:** 4

**Summary:**

This paper investigates a new integrated optimization method targeting at black-box optimization within high-dimensional spaces scenario, called Generative Evolutionary Optimization (GEO).
Different from the popular black-box optimization method - Bayesian optimization, GEO exhibits a linear time complexity.
Intrinsically, Geo a black-box optimization method that combines an evolutionary strategy with a generative surrogate neural network (GSN) model,
and the two basic components could function in a synergetic manner.
Specifically, evolutionary strategy helps to deal with the stability of the surrogate network training for GSN,
while GSN improves the mutation efficiency (sample efficiency) of the former.
Since the fitness results are combined and ranked using non-dominated sorting in GEO, it can be applied to multi-objective scenario.
Besides, the age evolution strategy is leveraged to dominated sorting step when the target function is stochastic.
Finally, the experimental findings reveal that GEO can accomplish the mentioned objectives: optimizing convex, high-dimensional, multi-objective, and stochastic target functions while maintaining O(N) complexity.


**Strengths:**

1. This paper is well-written and easy-to-follow, and the following parts are highlights: technique explanation, limitation analysis, high-level summary.
2. The technical design (GSN, ES, training stability) is reasonable, and the experimental evaluation is clear.
3. The key design in the cooperative framework is novel, which integrates the strengths of both GSN and ES while mitigating their weaknesses.


**Weaknesses:**

1. The specific parameter settings are not clear.
2. The reviewer suggests that in the discussion chapter, the related multi-objective high-dimensional solutions from Bayesian optimization community could be analyzed in terms of time complexity or efficiency if possible.
3. The test function in the experimental evaluation is limited, and this hamper the evaluation confidence. As mentioned by the authors, more test functions from different domains (maybe a fair benchmark) should be included to evaluate the performance of GEO.
In addition to Ackley, Rosenbrock and Styblinski-Tang, there are many objective function family, including CONSTR, SRN and so on.


**Questions:**

1. How many initial points are sampled in the latin hypercube stage.
2. What is the setting for the parameters in GEO, e.g., the number of generator. These greatly affect the reproducibility.

---

> ### Author Rebuttal · Authors · 2023-08-02
>
> Response to the Review:
>
> Thank you for your thorough review.
> First, here are the answers to the questions raised:
>
> Q1. How many initial points are sampled in the Latin hypercube stage?
>
> A1. The number is equal to the size of the generator pool used in the evolutionary strategy (ES). For example, if the pool size is 1000, the LHC initialization will also begin at 1000. We matched the initial point count for two types of initializations, random generator network initializations and Latin Hypercube Sampling (LHC). The maximum number is determined by the pool size when initializing generator network, hence the LHC initialization point count was matched to the pool size.
>
> Q2. What is the setting for the parameters in GEO, e.g., the number of generators? These greatly affect the reproducibility.
>
> A2. The number of generators corresponds to the pool size, which varies but was 1000 for the 2-objective problem in our experiments. Effects concerning the pool size can be found in the Supplement.
>
> Regarding other major parameters:
> Details of the neural network structure are also included in the Supplement paper. Although any kind of neural network can be used in GEO, we primarily adopted a Transformer-based structure for these experiments. Since it is a self-attention, structurally it is similar to GPT (Generative Pretrained Transformer). Without using the Trunk-branch trick, the Generator consists of six self-attention transformer blocks (designated as ASB in the Supplement figures). Though commonly GPT and other self-attentions use GeLU as activation, we employed HSWISH in these experiments. This was not for a particular reason but due to my misconception in 2021 that HSWISH was superior to GeLU. However, since the effect of activation is minimal, it was retained as is.
>
> In addition, I’d like to highlight that we have already found that GEO works with CNN-based networks. We refrained from including CNN-based experiments in the paper to avoid unnecessary confusion of readers. I want to emphasize that the type does not matter much as long as the neural network is sufficiently large.
>
> There are many other important parameters:
> - Boundary condition
> - Pool size
> - Trunk-branch trick
>
> Among these, the method of setting the boundary condition greatly influences the performance.
> These factors are described in the Supplement, and I kindly request you to refer to it for more information.
>
> ————————-
>
> My response to the weaknesses mentioned:
>
> W1.The specific parameter settings are not clear.
>
> A1. I believe that the Supplement provides a sufficient explanation of the specific parameters. There are additional experiments concerning pool size (= # of generators), boundary conditions, etc., and since the network structure is fundamentally similar to GPT, it can be easily reproduced. The Trunk-branch trick is also described in the Supplement.
>
> W2. The reviewer suggests that in the discussion chapter, the related multi-objective high-dimensional solutions from the Bayesian optimization community could be analyzed in terms of time complexity or efficiency if possible.
>
> A2. We often received questions comparing GEO with Bayesian optimization when presenting GEO, and we are aware of such demands.
> Bayesian optimizations are particularly effective in optimization, especially for the lower dimensions, whereas GEO demonstrates inefficiency in low dimensions and becomes efficient only in high-dimensional settings.
> Comparing these two algorithms with entirely different characteristics was challenging. If one performs better in a medium area (e.g., dimension = 20), readers may misunderstand that one is superior to the other. However, this is not the case.
> Bayesian optimizations and GEO (or conventional ES) are designed for different purposes, and as a result, grey areas inevitably arise. Comparisons within these grey areas can be quite ambiguous.
> Therefore, to prevent misunderstandings, we avoided direct comparisons.
>
> While I agree with your point somehow, I think it would be better to explore that idea in a separate study rather than including it in this paper.
>
> W3. The test function in the experimental evaluation is limited, and this hampers the evaluation confidence. As mentioned by the authors, more test functions from different domains (maybe a fair benchmark) should be included to evaluate the performance of GEO. In addition to Ackley, Rosenbrock, and Styblinski-Tang, there are many objective function families, including CONSTR, SRN, and so on.
>
> A3. I agree with this point.
>
> As I mentioned in the "global rebuttal," many more experiments were conducted in this research than what was included in the paper. However, unnecessary data was removed, considering issues such as figure readability.
>
> I am planning including the extra experiments in the Supplement. Even if new experiments are conducted, they are unlikely to significantly impact the main point of this study.
>
> Once again, thank you for your review.

---

> > ### Comment · Reviewer_7diD · 2023-08-12
> >
> > Thanks for your response, and we will carefully consider them in the following phase.

---

> > > ### Author Response · Authors · 2023-08-20
> > > **Thank you for your review**
> > >
> > > Thank you for your review

---

### Official Review · Reviewer_baKP · 2023-07-07

**Soundness:** 2 fair
**Presentation:** 2 fair
**Contribution:** 2 fair
**Rating:** 4
**Confidence:** 2

**Summary:**

In this paper, a black-box optimization approach is proposed that combines an evolutionary strategy (ES) with a generative surrogate neural network (GSN) model. This integrated model is designed to function in a complementary manner, where ES addresses the instability inherent in surrogate neural network learning associated with GSN models, and GSN improves the mutation efficiency of ES. From the overall view of this paper, the authors basically expressed clearly the point of innovation and the proposed algorithm.

**Strengths:**

The organization of this paper and the technical details of the proposed method are clear and easy to follow.


**Weaknesses:**

Theoretical derivations and proofs are lacking, and the validity of the method is difficult to be supported.

**Questions:**

1.In the introduction, there is no summary of the contribution, and I suggest that the contribution of this work be further emphasized.
2.This paper lacks experimental results and analysis of some arguments. For example, please explain why GEO is less efficient at lower sizes.
3.The writing of the paper could be improved for better description and clarification.
4.The selected algorithms for comparison are not new and are not state-of-the-art, please add the recently proposed multi-objective evolutionary approach for comparison.


**Limitations:**

The relevant limitations are described, but not in depth and specific enough.

---

> ### Author Rebuttal · Authors · 2023-08-04
>
> Thank you for your review. I will respond to each question one by one.
>
> Q1. In the introduction, there is no summary of the contribution, and I suggest that the contribution of this work be further emphasized.
>
> A1. This issue seems to be a writing problem similar to your 3rd question. In short, the core idea is that we've inherited the ideas from LGSO and EGAN to make a GSN-based optimization algorithm that functions much better in higher dimensions.
>
> We tended to elaborate on our words to convey the results about the performance as smoothly as possible, and to prevent it from appearing overly emphasized. This may have caused the summary to seem insufficient. Adding a brief summary to the introduction, as per your suggestion, would be a good idea.
>
> Q2. This paper lacks experimental results and analysis of some arguments. For example, please explain why GEO is less efficient at lower sizes.
>
> A2. This is an important question. The reduced performance of GEO in lower dimensions is due to GEO being based on neural networks.
>
> Neural networks show very high generalization performance when learning from a large amount of data, but when there's less data, performance sharply decreases due to overfitting. Particularly, this tendency is stronger with larger neural network sizes.
>
> Optimization in lower dimensions can be achieved with fewer data points (e.g., optimization in 2-dim. might be sufficiently done with 100 data points), so algorithms based on neural networks may show weakness in these situations.
>
> Also, we used a specific structure called a self-attention Transformer in GEO for this experiment (We explained the reason in Supplement). Transformers are known to be more suitable for large data but tend to perform poorly with small data. This trend made GEO's performance at lower dimensions get worse. Structurally, the decline in neural networks' performance with less data is an intrinsic problem that cannot be avoided. Hence, I emphasized in the discussion section that GEO is suitable for higher dimensions when there's a possibility for many target function evaluations.
>
> The characteristics of neural networks are generally well-known, so I thought that an explanation might not be necessary. However, based on your suggestion, it seems that a more comprehensive explanation would indeed be beneficial. Adding a discussion on this topic to the supplement appears to be a good approach. Thank you for your valuable insight.
>
> Q3. The writing of the paper could be improved for better description and clarification.
>
> A3. This is similar to Q1. A short summary of our research goals in the introduction would have made things clearer. Although the current version has been proofread and improved for reader comprehension, it appears you have pointed out that there are still improvements to be made in readability. I will review this further.
>
> Q4. The selected algorithms for comparison are not new and are not state-of-the-art, please add the recently proposed multi-objective evolutionary approach for comparison.
>
> A4.1. In fact, we conducted far more experiments than what has been included in the main text of the paper.
> We conducted the comparison experiments to the best of our ability during the initial stages of research, using available python packages. As is well-known, classical ES fails to optimize in higher dimensions, a fact that we confirmed (meaning that all the classical ES we tried failed to optimize at the target value of 8192 dimensions, so their differences were practically nonexistent). Therefore, we selected the most representative and well-known ES algorithms, intending to show concise and meaningful graphs. This approach may have inadvertently appeared as a lack of data.
>
> A.4.2. Additional answers about SOTA (State-of-the-Art):
>
> As I mentioned in the Discussion section of the main text and in the "global rebuttal," the  SOTA (or "best" ) of black-box optimization algorithm is not clear-cut. Even algorithms that claim superior performance may actually perform worse depending on hyperparameter settings and target functions. This occurs frequently, and particularly disappointing results often arise in real-world problems. Such considerations have brought about dilemmas in choosing between the latest algorithms and those that are well-known and widely used.
>
>
> Another reason:
>
> Even in LGSO, which is the most crucial paper for comparison, the testing was not conducted densely. Therefore, we thought that a similar amount of data would be sufficient.
> Additionally, the main objective of this paper was to overcome the limitations of the GSN algorithms, so we thought that we have successfully shown our goal.
>
> Despite this, more data is always beneficial, so we plan to upload additional materials to the Supplement.
> Thank you for your insightful comment.
>
> -------------
>
> Once again, thank you for your review.

---

> > ### Comment · Reviewer_baKP · 2023-08-15
> >
> > Dear authors: I extend my gratitude for your thorough responses and the inclusion of additional experiments. Overall, I have no more in-depth questions about this paper. Given the improvements in the revised version concerning the method description and empirical research, I am inclined to change my evaluation from "Borderline reject" to "Borderline accept".

---

> > > ### Author Response · Authors · 2023-08-20
> > > **Thank you for your review**
> > >
> > > Thank you for your review

---

### Official Review · Reviewer_GwEM · 2023-07-07

**Soundness:** 2 fair
**Presentation:** 2 fair
**Contribution:** 2 fair
**Rating:** 4
**Confidence:** 3

**Summary:**

The paper introduces a new method called Generative Evolutionary Optimization (GEO) that aims to address the challenges of black-box optimization in high-dimensional problems. The authors highlight that existing algorithms, such as Evolution Strategies (ES) and Bayesian optimization, have limitations when it comes to optimizing high-dimensional, non-convex problems while maintaining linear time complexity. They propose GEO as a combination of ES and Generative Surrogate Neural networks (GSNs) to achieve better performance in terms of stability, mutation efficiency, and optimization in high dimensions. The paper outlines the goals of GEO, discusses related works (L-GSO and EGAN), presents the methodology, and provides experimental results showing GEO's superiority over traditional ES and GSN in higher dimensions.

**Strengths:**

- The paper addresses an important problem in black-box optimization: optimizing high-dimensional, non-convex problems while maintaining linear time complexity
- The introduction provides a clear overview of the challenges faced by existing algorithms and the potential benefits of using GSN-based approaches like GEO
- The goals of GEO are well-defined, and the paper sets the stage for discussing the methodology and experimental results
- Combining EA with GSN is novel and interesting


**Weaknesses:**

- Some simple ES algorithms, such as OpenAI-ES [1], can optimize about 100k parameters in their paper; it is used to optimize the weight of the policy network. Although the idea of this paper seems novel and interesting, I am not sure that the 10k params can be called high-dimensional.
- The main results are shown in Figure 3, but it is unclear which function is used for 3-a), and the figure is not explained in the manuscript.

[1] Salimans, Tim, et al. "Evolution strategies as a scalable alternative to reinforcement learning." arXiv preprint arXiv:1703.03864 (2017).


**Questions:**

- I think the authors should provide more results compared with the previous methods similar to Figure 3-a), with various functions and algorithms. If there is no space for plotting all results, I think the authors should summarize them in a table. In the evosax [2], an open-source ES library, there is an example code for testing many ES algorithms in the ES benchmark function like Rosenbrock.

[2] https://github.com/RobertTLange/evosax/tree/main

**Limitations:**

- In the single-objective function, I think more algorithms should be compared to the proposed algorithm.

---

> ### Author Rebuttal · Authors · 2023-08-04
>
>
> Thank you for your review. I will address the Weaknesses and Question you pointed out one by one.
>
> ----------
>
> W1. Some simple ES algorithms, such as OpenAI-ES [1], can optimize about 100k parameters in their paper; it is used to optimize the weight of the policy network. Although the idea of this paper seems novel and interesting, I am not sure that the 10k params can be called high-dimensional.
>
> [1] Salimans, Tim, et al. "Evolution strategies as a scalable alternative to reinforcement learning." arXiv preprint arXiv:1703.03864 (2017).
>
> A1. We are aware of OpenAI-ES [1] and have cited it in the discussion section of our main text.
> OpenAI-ES targets the optimization of neural networks.
> However, neural networks differ significantly in nature from black-box optimization test functions. The characteristics of neural networks are more akin to those of convex functions.
>
> Though neural networks indeed consist of many parameters, almost all the points where the gradient is 0 are saddle points. It is contrary to the design of non-convex test functions in black-box optimization, which often have many local optima.
>
> Typical training methods like backpropagation gradient descent are more concerned with vanishing/exploding gradients rather than local optimum traps. If neural networks behaved like the Ackley function, with numerous local optima, expecting them to be optimized by gradient descent would be unrealistic.
>
> In other words, neural networks are generally closer in characteristics to convex functions, and thus the context of the research in "Evolution Strategies as a Scalable Alternative to Reinforcement Learning" is entirely different from typical black-box optimization.
>
> In most non-convex black-box optimization research, around 100 dimensions is considered high, and our experiment's extension to nearly 10,000 dimensions justifies calling it high-dimensional.
>
> The distinction between Non-convex and Convex (or semi-convex) is critical, as I formerly experienced that some researchers have misleadingly claimed state-of-the-art performance without proper delineation.
>
> W2. The main results are shown in Figure 3, but it is unclear which function is used for 3-a), and the figure is not explained in the manuscript.
>
> A2. The function used is Styblinski-Tang test function(with the ground state readjusted to 0).
> This seems to have been overlooked during the paper's revision. Thank you for pointing it out.
>
> Q. I think the authors should provide more results compared with the previous methods similar to Figure 3-a), with various functions and algorithms. If there is no space for plotting all results, I think the authors should summarize them in a table. In the evosax [2], an open-source ES library, there is an example code for testing many ES algorithms in the ES benchmark function like Rosenbrock.
>
> A. Additional experiments that were not presented in the main text are already included in the supplement. However, I acknowledge your question that this may still be insufficient, and I will answer accordingly.
>
> We conducted comparative experiments using a package very similar to evosax, called Pymoo, and a non-open package.
>  In fact, we conducted far more experiments than what has been included in the main text of the paper. As is well-known, classical ES fails to optimize in higher dimensions, a fact that we confirmed (meaning that all the classical ES we tried failed to optimize at the target value of 8192 dimensions, so their differences were practically nonexistent). Therefore, we selected the most representative and well-known ES algorithms, intending to show concise and meaningful graphs. This approach may have inadvertently appeared as a lack of data.
>
> Another reason:
>
> Even in LGSO, which is the most crucial paper for comparison, the testing was not conducted densely. Therefore, we thought that a similar amount of data would be sufficient. Additionally, the main objective of this paper was to overcome the limitations of the GSN algorithms, so we thought that we have successfully shown our goal.
>
> Despite the reasons mentioned above, it is true that the more experimental data we can show, the better.
> Choosing where to put extra data - in the main text or the supplement - needs a bit of thought. It'd be great to add it into the main text, but if we do, we might need to cut out some of the discussion or other data.
> So, I am considering upload extra data in Supplement.
>
> ---------
>
> Note 1.
>
> Although the Rosenbrock function is commonly used as a test function, it has a unique form. The Rosenbrock function is designed to test an algorithm's ability to search sharp valley region, rather than its capacity to escape many local optima.
> Therefore, I believe that functions such as Ackley or Styblinski-Tang would serve as more appropriate tests compared to the Rosenbrock function.)
>
> ------------
>
> Note 2.
>
> Following your suggestion, I recently conducted a straight forward test using the evosax package (6th Aug. W/ Rastrigin function). As anticipated, all of the algorithms ["SimpleES", "SimpleGA", "PSO", "DE", "Sep_CMA_ES", "Full_iAMaLGaM", "Indep_iAMaLGaM", "MA_ES", "LM_MA_ES", "RmES", "GLD", "SimAnneal", "GESMR_GA", "SAMR_GA"] exhibited a sharp decrease in performance as the dimensions increased. Particularly in a 1000-dimensional space, newly developed algorithms like GESMR_GA are showing poorer results than classical and well-validated algorithms such as Simple ES and PSO. (While changing the hyperparameter settings might result in different outcomes, the process of identifying the optimal parameters through repeated experiments cannot be regarded as black-box optimization.)
>
> This phenomenon is very common and is something I've previously encountered in my experiments. Due to these issues with generalization, we found that algorithms that are both newly developed and less well-known may actually be at a disadvantage in the validation process.
>
> -----------
>
> Once again, thank you for your diligent review.

---

### Author Rebuttal · Authors · 2023-08-04

Dear Reviewers,

I extend my heartfelt thanks for your diligent reviews. I have responded to each of your questions individually. However, given the character limit in the Rebuttal section, I was constrained to provide only brief answers to each question. Should you have any additional questions, I am ready and willing to offer more detailed responses.

----------
About Experimental Details & the Supplement document:

Many questions were raised regarding the details of the experiments. Due to the NeurIPS paper regulations limiting the content to 9 pages, I had to place the experimental details in the Supplement. In the Supplement, you will find information on the GEO network structure, additional experiments concerning key parameters, efforts to handle Transformers efficiently, more experiments on conventional test functions, discussions on non-dominated sorting, and so on. I kindly refer you to this section for further insights.

In the early version of this paper, I attempted to include all the details within the main text. However, due to the 9-page limit, the explanation of GEO's main concept was reduced, leading to feedback that the paper was hard to understand. Therefore, in this revision version, I have focused on including only the most critical parts in the main text, while boldly omitting or relocating less essential sections to the Supplement. I kindly request your understanding and leniency regarding this matter.

---------------
About Additional Experiments:

From the initial stages of our research, we conducted a far greater number of experiments than what is included in this paper. However, for the sake of readability, we retained only the experiments we considered most important and excluded the rest. It seems that our attempt has unfortunately been perceived as a lack of data.

The specific situation is as follows:

We carried out as many experiments as possible on ES using Python packages. It is a well-known fact that classical ES does not perform well in high-dimensional optimization, our experimental results were also consistent with that. The most crucial experiment in this paper is the performance comparison at 8192 dimensions, where all the classical ES we tried failed to optimize. Since classical ES can't optimize in such high dimensions, we felt that the comparison would be meaningless and too much data points would impair the readability of the graph. Therefore, we only kept the most renowned and widely-used algorithms.

(In addition to conventional test functions, we have extensively carried out simulations with practical environments such as electronic circuit simulators. However, these were not included in the paper to prevent potential reader confusion and due to the confidentiality terms associated with the provided simulators.)

Meanwhile, our focus was more on the GSN perspective. Therefore, we believed that demonstrating that GEO surpasses LGSO was a sufficient explanation. Also, I would like to note that the superior performance of LGSO compared to traditional ES has already been demonstrated in LGSO's own research.

While we felt we had a good reason for excluding the unnecessary data, but it raised a lot of questions in this revision. Therefore, we plan to reincorporate the additional experiments, and due to space limitations in the main text, we believe that adding them to the supplement would be more appropriate.

----------------
About SOTA:

When referring to the "state-of-the-art" (SOTA) in black-box optimization algorithms, we must be extra careful. This is especially true when compared to other machine learning fields.

While the term "SOTA" often implies "the best", at least in this field, it cannot be understood so simply.
The reason for this is that optimization algorithms vary widely in their goals and the conditions under which they work well. Hyperparameter settings, test functions, and the algorithm itself are strongly correlated, leading to situations where an algorithm that seems to be the best may perform poorly in specific scenarios, and vice versa. And such a situation can occur quite frequently.

From this perspective, our experiments were not intended to demonstrate that GEO outperforms other optimization methods in every scenario. Rather, we aimed to highlight that, unlike traditional methods which suffer from the "Curse of Dimensionality", GEO does not lose as much effectiveness as the number of dimensions increases. Also, this does not imply that GEO is always the ideal choice or a perfect solution. Every optimizer has its strengths and weaknesses depending on the specific situation, and there may be unpredictable circumstances where GEO does not perform as well.

Therefore, while our findings, including those from experiments we haven't disclosed, consistently show GEO outperforming other algorithms, we're careful not to quickly say that GEO is the best solution.

We hope our intention has been conveyed without any misunderstanding.

------------

Note.

Following a suggestion from one of the reviewers, I recently conducted a straight forward test using the evosax package (6th Aug., W/ Rastrigin function). As anticipated, all of the algorithms ["SimpleES", "SimpleGA", "PSO", "DE", "Sep_CMA_ES", "Full_iAMaLGaM", "Indep_iAMaLGaM", "MA_ES", "LM_MA_ES", "RmES", "GLD", "SimAnneal", "GESMR_GA", "SAMR_GA"] exhibited a sharp decrease in performance as the dimensions increased. Particularly in a 1000-dimensional space, newly developed algorithms like GESMR_GA are showing poorer results than classical and well-validated algorithms such as Simple ES and PSO.

This phenomenon is very common and is something I've previously encountered in my experiments. Due to these issues with generalization, we found that algorithms that are both newly developed and less well-known may actually be at a disadvantage in the validation process.

-------------

Once again, thank you to the reviewers. I look forward to your further feedback.

---

### Decision · Program_Chairs · 2023-09-21

**Decision:**

Reject

**Comment:**

The paper studies evolutionary algorithms with generative surrogates for black-box optimization. The reviews were generally borderline with primarily three concerns. The first concern was regarding clarity, as many details on the experiments were omitted. This part is relatively fixable with a better organization and structuring of the paper. The second concern was regarding the evaluation of the paper. The paper looks at many synthetic functions, but it is unclear how well do these observations translate to real-world domains. One suggestion in this regard is to look at some real-world benchmarks such as Design-Bench [1] and NAS-Bench. The third concern is with the scoping of the paper. While the reviewers found some evidence that the work presents an improvement over existing EO algorithms, the paper conflates these improvements over relatively weak/limited BO baselines. For instance, there is a good body of work showing the use of generative models for black-box optimization e.g., see [2,3] and the references therein. These methods are not discussed and compared against, giving an incomplete picture of how this work fits in with those prior works both conceptually and empirically.


[1] Design-Bench: Benchmarks for Data-Driven Offline Model-Based Optimization. https://arxiv.org/abs/2202.08450
[2] Model Inversion Networks for Model-Based Optimization. https://proceedings.neurips.cc/paper/2020/hash/373e4c5d8edfa8b74fd4b6791d0cf6dc-Abstract.html
[3] Generative Pretraining for Black-Box Optimization. https://arxiv.org/abs/2206.10786